# Therapeutic Values of Myeloid-Derived Suppressor Cells in Hepatocellular Carcinoma: Facts and Hopes

**DOI:** 10.3390/cancers13205127

**Published:** 2021-10-13

**Authors:** Yijun Wang, Tongyue Zhang, Mengyu Sun, Xiaoyu Ji, Meng Xie, Wenjie Huang, Limin Xia

**Affiliations:** 1Hubei Key Laboratory of Hepato-Pancreato-Biliary Diseases, Department of Gastroenterology, Institute of Liver and Gastrointestinal Diseases, Tongji Hospital of Tongji Medical College, Huazhong University of Science and Technology, Wuhan 430030, China; M202076163@hust.edu.cn (Y.W.); M201976049@hust.edu.cn (T.Z.); sunmengyu@stu.ahmu.edu.cn (M.S.); d202081875@hust.edu.cn (X.J.); xiemeng@hust.edu.cn (M.X.); 2Hepatic Surgery Center, Hubei Key Laboratory of Hepato-Pancreato-Biliary Diseases, Tongji Hospital of Tongji Medical College, Huazhong University of Science and Technology, Wuhan 430030, China

**Keywords:** myeloid-derived suppressor cells, hepatocellular carcinoma, HCC, combination therapy, immunotherapy, tumor microenvironment, immune checkpoint inhibitor, biomarker

## Abstract

**Simple Summary:**

Myeloid-derived suppressor cells restrict the effectiveness of immune-checkpoint inhibitors for a subset of patients mainly through thwarting T cell infiltration into tumor sites. Treatments targeting MDSCs have shown potent inhibitory effects on multiple tumors, including hepatocellular carcinoma. In this review, we summarize the pathological mechanisms of MDSCs and their clinical significance as prognostic and predictive biomarkers for HCC patients, and we provide the latest progress of MDSCs-targeting treatment in HCC.

**Abstract:**

One of the major challenges in hepatocellular carcinoma (HCC) treatment is drug resistance and low responsiveness to systemic therapies, partly due to insufficient T cell infiltration. Myeloid-derived suppressor cells (MDSCs) are immature marrow-derived cell populations with heterogeneity and immunosuppression characteristics and are essential components of the suppressive tumor immune microenvironment (TIME). Increasing evidence has demonstrated that MDSCs are indispensable contributing factors to HCC development in a T cell-dependent or non-dependent manner. Clinically, the frequency of MDSCs is firmly linked to HCC clinical outcomes and the effectiveness of immune checkpoint inhibitors (ICIs) and tyrosine kinase inhibitors (TKIs). Furthermore, MDSCs can also be used as prognostic and predictive biomarkers for patients with HCC. Therefore, treatments reprograming MDSCs may offer potential therapeutic opportunities in HCC. Here, we recapitulated the dynamic relevance of MDSCs in the initiation and development of HCC and paid special attention to the effect of MDSCs on T cells infiltration in HCC. Finally, we pointed out the potential therapeutic effect of targeting MDSCs alone or in combination, hoping to provide new insights into HCC treatment.

## 1. Introduction

Primary liver cancer ranked as the sixth most commonly diagnosed cancer and the third leading cause of cancer death globally in 2020, with approximately 906,000 new cases and 830,000 deaths [1]. HCC accounts for more than 80% of primary liver cancer and is the most common primary liver cancer [1,2]. Due to the hidden nature of HCC in the early stage, many HCC patients have progressed into the middle, late, or even terminal stage when they are diagnosed and thus miss the optimal curative treatment [2]. Although the appearance of sorafenib and subsequent TKIs has brought a glimmer of hope for advanced and end-stage patients, the therapeutic effects are not optimistic [3,4,5]. In recent years, immunotherapy, especially immune checkpoint inhibitors represented by PD-1/PD-L1, CTLA-4, etc., have revolutionized the landscape of therapy in several advanced tumors [6]. However, a significant proportion of HCC patients still cannot benefit from this treatment. Some patients even develop resistance after the initial response, possibly due to the suppressive TIME in HCC [7]. More and more combination approaches are now being attempted to address ICIs monotherapy resistance [8]. Up to the present, the combination of ICIs with atezolizumab and VEGF inhibitor with bevacizumab is the first-line treatment for patients with advanced HCC unless contraindications [9,10]. In addition, other immune adjuvant therapies, such as drugs that target the immunosuppressive microenvironment within the tumor, also improve the response to immunotherapy.

Regulatory T cells (Treg), M2-polarized tumor-associated macrophages (TAM2), and MDSCs are TIME’s primary types of immunosuppressive cells. The accumulation of these cells set the limitation of antitumor efficiency and led to the initiation and progression of HCC by establishing an immunosuppressive environment [7,11]. Researchers have found that targeting these cells can assist in reprogramming immune microenvironments [12,13,14]. MDSCs, as the hub node in these suppressor cells, connect a variety of immune cells, stromal cells, and tumor cells in the process of tumorigenesis. A series of unique intrinsic signals are abnormally activated during HCC development, such as cell cycle-related kinase (CCRK) signaling. CCRK signaling causes strong immunosuppression through the activation, recruitment, and accumulation of MDSCs, leading to anti-immune checkpoint blocking resistance [15]. In this review, we summarize how MDSCs are activated and recruited into HCC and how they promote the initiation and development of HCC. In addition, we recapitulate information about current treatments about MDSCs in HCC and potentially propose effective combined treatment strategies.

## 2. The Nature of MDSCs

Myelopoiesis is a common process in healthy bodies. The transition from hematopoietic stem cells to myeloid precursors to mature immune cells is strictly regulated [16]. When an acute infection or trauma occurs, massive immature myeloid cells (IMCs) are released from the bone marrow and quickly differentiate into mature myeloid cells, such as polymorphonuclear neutrophils and monocytes, helping the body eliminate these acute pathological conditions [17]. However, in patients with tumors or chronic inflammation, continuous stimulation often leads to defective differentiation of IMCs. Such effector cells tend to secrete high levels of arginase 1 (ARG1), reactive oxygen (ROS), myeloperoxidase (MPO), nitric oxide (NO), and other products to exert immunosuppressive effects [16,17,18]. Considering the myeloid origin of these cells and their potent immunosuppressive activity, they are named myeloid-derived suppressor cells [18].

MDSCs are a highly heterogeneous population. There are currently two main subsets of MDSCs based on phenotypic and morphological features: monocytic MDSCs (M-MDSCs) and polymorphonuclear MDSCs (PMN-MDSCs, also known as G-MDSCs), which are similar to monocytes and neutrophils, respectively [17]. The phenotypes of mouse M-MDSCs and PMN-MDSCs are different from those of humans. Generally, mouse MDSCs are characterized by myeloid differentiation markers: CD11b^+^ and Gr-1^+^ (Ly6G^+^ or Ly6C^+^). Among them, PMN-MDSCs are defined as CD11b^+^Ly6C^low^Ly6G^+^, while M-MDSCs are defined as CD11b^+^Ly6C^high^Ly6G^−^ [18]. However, Gr-1 is not expressed on human leukocytes. It has taken several years for scientists to find out the specific markers that distinguish these two subgroups in humans [19]. It is now generally accepted that human MDSCs are commonly marked as CD11b^+^, CD33^+^, and HLA-DR^−^. Considering that CD14^+^ and CD15^+^ or CD66b^+^ are the markers of monocyte lineage and granulocyte, respectively, PMN-MDSCs are defined as CD11b^+^CD14^−^CD15^+^ or CD11b^+^CD14^−^CD66b^+^, and M-MDSCs are defined as CD11b^+^CD14^+^HLA-DR^−^ [19]. Lectin-type oxidized LDL receptor 1 (LOX-1) is a novel marker of PMN-MDSCs, expected to distinguish PMN-MDSCs from neutrophils [20]. Nan et al., observed the elevation levels of LOX-1^+^ CD15^+^ PMN-MDSCs in HCC patients compared with healthy control and patients with chronic hepatitis B [21]. Additionally, the circulation LOX-1^+^ CD15^+^ PMN-MDSCs were positively associated with those in HCC tissues [21]. Except for these two common subsets, there is a third type named early MDSCs (eMDSCs). This subtype contains more immature progenitors, which has colony formation and the function of differentiating into MDSCs and other myeloid precursor cells [16]. Currently, there havee been no data available about the role of eMDSCs in HCC.

A prominent feature of MDSCs is immunosuppression. The distribution and functions of the two MDSCs are different. Generally, PMN-MDSCs are more abundant in the tumor microenvironment, while M-MDSCs accumulate in peripheral blood and show a more substantial inhibitory effect [22,23,24,25]. M-MDSCs express high levels of iNOS but release low ROS. They mainly rely on producing substances with immunosuppressive properties, such as NO, and suppress the immune response less selectively. In contrast, PMN-MDSCs show a weaker chemotactic response and prefer to produce NADPH oxidase (NOX2) and reactive oxygen species (ROS) to mediate tolerance of T cells and the antigen-specific suppression [11]. There are various mechanisms for MDSCs to suppress the immune response, but these mechanisms do not function simultaneously. The specific inhibitory effect depends on the disease location, type, progression stage, and the subtype of MDSCs clustering in the lesion [26].

In addition to immunosuppression, MDSCs are also highly plastic in tumors. MDSCs can transform into various immune cells such as TAM, dendritic cells (DCs), mature macrophages, and neutrophils in different tumor microenvironments and perform diverse functions mediated by multiple growth factors and cytokines from tumor microenvironment or chronic inflammation [27].

## 3. The Accumulation of MDSCs in HCC and Its Clinical Significance

In 2013, Tamar Kapanadze et al., have found the expansion of MDSCs in all four HCC mouse models (subcutaneous implantation, in-situ implantation, DEN-induction, MYC-ON-induction) [28]. MDSCs increased in both transplantation models. While in the other two models, there was no significant increase until the tumor reached the advanced stage [28]. The author also noted that the inhibitory effect of DEN-induced MDSCs was lower than that of the subcutaneous groups, attributed to the proportion difference of M-MDSCs and PMN-MDSCs [28]. Except in mice models, MDSCs also accumulated in HCC patients [29,30]. It is worth mentioning that most published experiments measured the concentration of MDSCs in the peripheral blood instead of tumor sites of HCC. Extensive evidence has also noticed that the abundance of MDSCs could be used as an independent prognostic and predictive factor for human HCC. A 2016 meta-analysis showed that patients with high levels of MDSCs tended to have poorer overall survival [31]. The frequency of MDSCs in peripheral blood was linearly regressed to tumor volume and moderately positively correlated to AFP [32]. In human HCC, the frequency of MDSCs was also closely linked to antitumor effects and tumor recurrence after liver transplantation [33]. Sorafenib exhibited a more potent therapeutic effect in mice with subcutaneous transplantation than those with orthotopic transplantation. That difference might be due to the increased MDSCs in orthotopic implanted tumors treated by sorafenib, which induced immunosuppression by inhibiting T cells function, thereby weakening the therapeutic effect of sorafenib [33]. Inhibition of MDSCs, with anti-ly6G or anti-IL-6 body, significantly improved the efficacy of sorafenib [33]. In addition, MDSCs were also involved in the resistance of HCC to anti-PD-L1 therapy. A study found that the percentage of PD-L1^+^MDSCs in the peripheral blood of HCC patients was much higher than that of healthy subjects. Therefore, the combination of anti-PD-L1 and MDSCs therapy might play a positive role [34]. Hepatic arterial infusion chemotherapy (HAIC) is an alternative therapy to sorafenib for advanced HCC. The HAIC treatment efficiency of patients with high pre-treatment MDSCs levels was worse, and their overall survival was shortened [35]. A study recently found that M-MDSCs, not PMN-MDSCs, were attracted to HCC and amplified through CXCL10/TLR4/MMP14 signaling, promoting tumor recurrence after liver transplantation [36]. The above information suggests that MDSCs are closely associated with the prognosis and treatment efficiency of HCC and may be an independent prognostic and predictor of HCC.

## 4. The Generation, Activation, and Recruitment of MDSCs in HCC

The generation and activation of MDSCs is a complicated process involving various tumor-derived cytokines and growth factors signals. Dmitry I. Gabrilovich has proposed to divide the generation and activation of MDSCs into two different but partially overlapping processes [37]. The first signal controls IMCs production, mainly mediated by tumor-derived growth factors, such as IL-6, IL-11, IL-17A, G-CSF, GM-CSF, TNFα, and involves signal pathways including STAT3, IRF8, C/EBPβ, RB1, Notch, adenosine receptors A2b, NLRP3, and others [38,39,40]. These signals stimulate myelopoiesis to produce massive IMCs and inhibit their differentiation into mature cells. The second signal then enables these IMCs to differentiate into immunosuppressive MDSCs. In this process, tumor necrosis factor-α (TNF-α), multiple interleukins (IL), prostaglandin E2 (PGE2) and cyclooxygenase 2 (COX2), and other pro-inflammatory factors derived from tumor stroma activate the intracellular signal, in which NF-κB is the most dominant signaling pathway during this process [17,39]. The Stat1, Stat6, and ER stress pathways are also reported to participate in this process [39]. These activated MDSCs and IMCs then mobilize from bone marrow or peripheral lymphoid tissue into blood circulation and are recruited to the target tumor sites at the chemotaxis of cytokines, complements, and chemokines.

The activation and recruitment mechanisms of MDSCs are similar and unique in the tumor microenvironments of HCC compared to other tumors. Multiple chemotactic factors secreted by the tumor microenvironment stimulate the activation and chemotaxis of MDSCs into the HCC (Table 1). Similar to other tumors, chemokines are the most prominent factors that meditate the migration of MDSCs. CCL2, CCL7, CCL9, CCL21, C-X-C motif ligand (CXCL)1, CXCL5, and CXCL12(SDF-1a) have been proven to participate in the development and migration of MDSCs through binding with the corresponding receptors in HCC [22,41,42,43,44]. A recent study found that complement C3 played a role in MDSCs [45]. When fatty acid metabolism was induced in HCC, a large amount of ROS generated from the tumor cells caused the secretion and activation of complement C3, which activated MDSCs and produced IL-10 through the P38 MAPK pathway to inhibit CD8^+^ T cells [45]. In addition to tumor cells, stromal cells in HCC also contributed to the generation and migration of MDSCs. Hepatic stellate cells (HSCs) are unique stromal cells in the liver microenvironment, which secrete various cytokines and inflammatory-related molecules to regulate the immune response in HCC [43,46,47]. Both hepatoma cells and HSCs secreted IL-6 and CXCL12 to promote the generation and expansion of MDSCs in HCC, thereby stimulating the progression of HCC [42,47,48]. The expression levels of IL-1α, IL-1β, and IL-6 were positively correlated with the proportion of MDSCs and Foxp3^+^Treg in HCC [48]. HSCs-derived PGE2 and COX2 also contributed to the accumulation of MDSCs in tumor sites and Treg [46]. Tumor-associated fibroblasts (TAFs) mediated monocytes’ migration into tumor sites by secreting CXCL12 and induced their transformation into CD14+HLA-DR-/low MDSCs by IL-6 mediated STAT3 activation.

## 5. The Mechanisms of MDSCs Promoting HCC

MDSCs executed potent immunosuppressive effects through various mechanisms, ultimately inhibiting the immune effector cells (CD8^+^ T cells, NK cells, etc.) or inducing immunosuppressive cells (Treg and TAM2) (Figure 1). However, these mechanisms are redundant, and they do not work concurrently. The specific mechanisms are related to the type and progression of the tumor [24].

### 5.1. Nutrient Depletion

The immunosuppression induced by MDSCs mainly depends on the activity of two enzymes: ARG1 and iNOS, which are highly expressed in MDSCs. L-arginine, for example, is a common substrate for iNOS and ARG1. As a conditionally essential amino acid for T cells, the abundance of L-arginine is closely related to T cells proliferation and differentiation [53,54]. However, due to the massive expansion of MDSCs in HCC-bearing mice and cancer patients, substantial secretion of iNOS and ARG1 are produced, which, in turn, causes the depletion of L-arginine. In fact, L-arginine is the most strongly depleted amino acid in the tumor microenvironment, and it is about five times lower in the tumors than in the periphery [55,56].

Mechanically, the depletion of L-arginine decreases levels of CD3 ζ-chain on T cells, impairing the assemble and stabilization of the TCR-CD3 complex [57]. The modified TCR-CD3 complex’s antigen recognition capability is weakened, as well as the following tumor antigen-specific immune responses. In addition, L-arginine starvation harms the formation of immune synapses between T cells and antigen-presenting cells (APC) through impeding the dephosphorylation of actin-binding protein cofilin [58]. These mechanisms down-regulate the response of T cells to antigen-specific signals. L-arginine also participates in the cell cycling of T cells. The absence of L-arginine arrests human T cells in the G0-G1 phase through silencing cyclin D3 and cyclin-dependent kinase 4 [59]. Moreover, L-arginine starvation leads to decreased retinoblastoma protein (Rb) phosphorylation, which is harmful for G2-S transition associated transcriptional genes [59]. In short, MDSCs interfere with T cells membrane signal transduction and T cells cycle by consuming nutrients needed for lymphocyte-activating proteins synthesis, thereby exerting a powerful inhibitory effect on T cells.

In human HCC, the expression level and enzyme activity of ARG1 expressed in PMN-MDSCs are observed to increase significantly in PBMC and tumor sites [47,60]. LOX-1^+^CD15^+^ PMN-MDSCs from HCC patients suppressed antigen non-specific T cells proliferation and IFNγ production, which can be reversed by ROS inhibitor NAC, ARG1 inhibitor, and L-arginine [21]. Additionally, the arginase activity in CD14^+^HLA-DR^−/low^ cells from HCC patients is two-fold higher than CD14^+^HLA^−^DR^+^ cells from healthy controls, and the addition of L-arginine or depletion of MDSC resulted in enhanced IFNγ secretion [60]. HSC-induced MDSCs also highly expressed ARG1 and iNOS to exert potent inhibitory T cells immune responses in HCC [47]. Moreover, higher levels of Arg1, Cyclooxygenase 2 (Cox2), and iNOS are expressed in chemotherapy-resistant HCC cells, mediating the immunosuppressive activity of MDSCs, and lead to low response of HCC to chemotherapy [61]. In general, this evidence indicates that L-arginine deprivation is one of the chief mechanisms of MDSCs in promoting HCC.

### 5.2. Oxidative Stress

ROS and reactive nitrogen species (RNS) are also major mechanisms involved in MDSCs-mediated immunosuppression. The fostered ROS production by MDSC is mediated by upregulating activity of NADPH oxidase 2(NOX2). ROS exerts toxic effects on most T cells by damaging proteins, lipids, and nucleic acids. However, MDSCs survive in excessive ROS environments, while T cells cannot [62]. H_2_O_2_ has been shown to suppress the adaptive immune response by decreasing the CD3-ζ chain expression, preventing T cells activation and IFNγ expression [63].

In addition to ROS, MDSCs produce high levels of RNS, predominantly NO, via the activation of iNOS. Although low levels of NO generated by CD8^+^ T cells are conducive to immune signal transduction, excessive NO from MDSCs leads to an opposite outcome [64]. High levels of NO induce nitration of CD3 ζ-chain and TCR-CD8 complex, leading to the dissociation of the TCR complex and disruption of antigen-specific recognition [65,66,67]. Additionally, NO nitrifies chemokine such as CXCL12, CCL21, CCL2, or CCL5, reducing chemokine-induced T cells migration and tumor infiltration [66,68]. Excessive NO also down-regulates the expression of the L-selectin lymph node homing receptor on CD8^+^ T, hindering homing and antigen-dependent activation of CD8^+^ cells in lymph nodes [69,70]. Finally, excessive NO induces the expression COX-2 and PGE, which increases several MDSCs-associated immunosuppressive factors, such as indoleamine2,3-dioxygenase1 (IDO), IL-10, ARG1, and others [26].

### 5.3. Induction of Immunosuppressive Cells

MDSCs can promote the clonal expansion of antigen-specific natural Treg cells and induce the conversion of naive CD4^+^ T cells into induced Treg cells. Bastian Hoechst et al., showed that CD14^+^HLA-DR^−/low^ cells from HCC patients could induce the expression of CD4^+^CD25^+^Foxp3^+^ regular T cells [60]. Treg cells generate IL-10 and TGF-b1 and exhibit a highly immunosuppressive phenotype in advanced HCC [29]. The author also noted a group of T cells that expressed a lower frequency of granzyme B in advanced HCC patients, and the elimination of Treg could restore the production of granzyme B [29]. Thus, it is assumed that Treg may thwart CD8^+^ T cells at least partly by inhibiting granzyme B production in effector CD8^+^ T cells [29]. The foster of granzyme B was correlated with prolonged progression-free survival after the combination of rituximab and chemotherapy in patients with follicular lymphoma [71]. In short, MDSCs indirectly enhance the tumor’s immuno-tolerance via driving the expansion of Treg cells in HCC.

### 5.4. MDSCs Suppress NK Cells and Kupffer Cells

MDSCs from patients with HCC inhibit NK cell cytotoxicity and cytokine secretion when cultured together in vitro. This suppression is dependent on cell contact, and NKp30 mediates contacts on NK cells rather than the ARG1 activity of MDSCs [72]. Kupffer cells represent the first line of defense against tumor cells in the liver. MDSCs are observed to alter the expression of the co-stimulatory/co-inhibitory molecules from Kupffer cells, such as CCL2, IL-10, IL-18, IL-6, and IL-1β [73]. Furthermore, MDSCs decrease antigen-presentation activity of Kupffer cells by promoting PD-L1 expression while reducing CD86 and MHCII expression on the Kupffer cells membrane [73].

### 5.5. Other Factors

Many other factors are involved in the immunosuppressive function of MDSCs, such as PD-L1, HIF-1α, and galectin 9. The percentage of PD-L1^+^ MDSCs in PBMCs from HCC patients is much higher than that of healthy donors and patients after treatment [34]. Moreover, the disease-free survival time of HCC patients is closely associated with PD-L1^+^MDSCs [34]. Galectin 9, a widely expressed soluble membrane molecule, is upregulated on MDSCs. Galectin 9 binds to T cells immunoglobulin and mucin domain-containing protein 3 (TIM3) on lymphocytes and induces T cells apoptosis [74]. Adisintegrin and metalloproteinase 17 (ADAM17) is a membrane molecular expressed on MDSCs. Through binding with CD62 ligand on CD4^+^ and CD8^+^ T cells, ADAM17 mediates the disruption of T cells homing [70]. In HCC, HIF-1α facilitates the dephosphorylation of extracellular ATP into 5′-AMP by ectonucleoside triphosphate diphosphohydrolase 2 (ENTPD2). Increasing 5′-AMP prevents M-MDSCs from maturation at tumor sites [50].

## 6. The Role of MDSCs in the Progression of HCC-Related Liver Diseases

The main risk factors for HCC are chronic hepatitis B virus (HBV) or hepatitis C virus (HCV) infection, excessive alcohol consumption, diabetes, and non-alcoholic fatty liver disease (NAFLD) [75]. An increasing number of studies have found that MDSCs play an important role in the pathogenesis of liver inflammation or cirrhosis and their progression to HCC. On the one hand, MDSCs may limit immune response and the subsequent tissue injury. While they may also favor the persistence of the virus in the liver by restricting the interference of T cell activity, causing chronic hepatitis and even HCC.

Hepatitis virus infection is the most prominent risk factor for HCC development [76]. There is a significant correlation between MDSCs levels and HBV/HCV disease progression and their response to antiviral therapies. The accumulation of MDSCs during chronic HBV or HCV infection may promote and sustain persistent virus infection. It was detected that the MDSCs with CD14^+^HLA-DR^−/low^ in the peripheral blood of HBV patients were significantly higher than those in healthy controls [77]. HBV promoted MDSCs differentiation via IL-6/ ERK/IL-6/STAT3, which, in turn, maintained HBV persistence and immunosuppression [77]. Another study observed that PMN-MDSCs were elevated in patients with HBV replication without liver damage (immunotolerance phase) [78]. Expanded

PMN-MDSCs suppressed T cell response partially via overexpressing ARG1, thereby sus- taining the state of immunotolerance to high levels of HBV replication [78].

Similarly, a higher percentage of MDSCs, defined as HLA-DR^−/low^CD11b^+^CD33^+^CD14^+^, has also been detected in the peripheral blood of HCV-infected patients compared with healthy controls [79]. The levels of MDSCs in HCV patients were positively correlated with the HCV viral load and the enzymes related to liver injury [79]. Moreover, HCV promoted the accumulation and differentiation of CD33^+^ MDSCs, causing viral persistence and anti-HCV vaccine non-responsiveness [80,81]. Thus, targeting MDSCs may restore T cell response to HBV and HCV by regulating the immune network and reducing viral load.

An increased percentage of MDSCs has also been detected in patients with alcoholic liver disease (ALD) or NAFLD. ALD includes multiple liver disease stages, including alcoholic steatosis, alcoholic hepatitis, alcoholic cirrhosis (ALC), and alcohol-related HCC [82]. Compared with alcoholic steatosis patients or normal people, ALC patients showed higher levels of PMN-MDSCs in peripheral blood and liver, and stronger inhibitory activity against NK cells [82]. NAFLD is the fastest-rising cause of HCC globally and the leading cause of HCC in the absence of cirrhosis [83]. Similarly, the frequency of CD11b+Gr1+ MDSCs in the liver of NAFLD mice was significantly higher than that of the control group, which may partly recruit to the NAFLD liver through the CCL2-CCR2 pathway and expanded through CSF stimulation [84]. These data suggest that MDSCs may be involved in the progression of liver cirrhosis exacerbation and is closely associated with HCC initiation. However, studies on the role of MDSCs in ALC or NAFLD progression are limited and remain to be further investigated.

## 7. Treatment Targeting MDSCs

As discussed above, numerous pieces of evidence support a close association between MDSCs accumulation and clinical outcomes and treatment effects of HCC. Therefore, targeting MDSCs has been extensively studied in many preclinical experiments, and some of these drugs have entered clinical trials. These strategies are carried out based on four main goals: (a) inhibit the generation of MDSCs and induce their apoptosis; (b) prevent the trafficking of MDSCs; (c) inhibit the immunosuppressive function of MDSCs; and (d) induce the differentiation and maturation of MDSCs (Figure 2).

### 7.1. Inhibition of MDSCs Generation and Induction of Their Apoptosis

As mentioned above, there are many stimulating factors causing the activation of signaling pathways in MDSCs precursor cells. Blocking these signaling pathways and critical stimulating factors might inhibit the generation of MDSCs.

Firstly, depletion of MDSCs could be achieved by the application of classical chemotherapeutics. The role of chemotherapeutic agents on immune responses is complicated. Some agents have been found to eliminate MDSCs in circulation and tumors, such as gemcitabine (Gem), 5-fluorouracil (5-FU), oxaliplatin, and capecitabine [85,86,87,88]. However, most HCC patients are not sensitive to chemotherapy due to drug resistance. Excessive chemotherapy may even inhibit the immune system, leading to adverse consequences such as liver parenchyma injury, reduced compensatory function of cirrhosis, and tumor progression [89].

Researchers are currently working on optimizing chemotherapeutics to achieve low-dose, low-toxicity, and high-efficiency effects. For example, L. Ringgaard. et al. have designed a liposomal oxaliplatin formulation (PCL8-U75), which depleted MDSCs more efficiently and inhibited tumor growth more profoundly [90]. Yuan Zhang. et al. have formulated gemcitabine monophosphate (GMP) into lipid-coated calcium phosphate (LCP) nanoparticles, namely GMP-LCP, which also showed superior inhibition on the MDSCs [86]. If these optimized chemotherapy drugs could achieve better targeting and effective absorption, they will provide essential options for the treatment of HCC. The combination of chemotherapy and PD-1/PD-L1 inhibitors has also exhibited antitumor effects in preclinical models and clinical trials. A phase II study (NCT03092895) has enrolled patients with advanced primary liver cancer to access the safety and efficiency of SHR-1210 (anti-PD-1 antibody) in combination with chemotherapy (FOLFOX4 and GEMOX regimen) (Table 2). The study noticed that the combination of SHR-1210 with FOLFOX4 and GEMOX regimen showed manageable toxicity and promising antitumor activity in patients with advanced HCC [91]. The combination of gemcitabine and nivolumab (anti-PD-1 antibody) are also included in the plans for clinical trials on non-small cell lung cancer (NSCLC), which are expected to enhance the immune checkpoint inhibitors by eliminating MDSCs (NCT04331626). Patrick Dillon et al. are now preparing a phase I clinical trial to assess the impact of focused ultrasound ablation, low-dose gemcitabine, or their combination on MDSC and T cells (NCT04796220).

TNF-related apoptosis-induced ligand receptors (TRAIL-Rs) are members of the TNF receptor superfamily [92]. It is expressed on the surface of MDSCs and causes MDSCs a shorter lifetime and higher apoptosis rates than neutrophils and monocytes. Thus, targeting TRAIL-Rs by selective agonists might be an option for cancer therapy by reducing the population of MDSCs [93]. DS-8273a was an agonist for TRAIL receptor 2 and was tested in a phase I trial (NCT02076451) in 16 advanced cancer patients, including patients with HCC [94]. This trial found that the application of DS-8273a could reduce the number of MDSCs in the peripheral blood of most patients to the levels of healthy volunteers temporarily while showing no effect on the number of neutrophils, monocytes, and other populations of myeloid and lymphoid cells. The decrease in MDSCs was correlated with the length of progression-free survival and provided an opportunity for patients to enhance the effect of cancer therapeutics [94].

STAT3 is a critical transcription factor for MDSCs expansion and immunosuppressive activity. Prajna Guha et al. have noted that STAT3 inhibitors (STATTIC or BBI608) dramatically decreased the accumulation of liver-associated MDSCs in tumor-bearing mice, leading to effective antitumor activity. Mechanically, STAT3 inhibitors induced the apoptosis of liver-associated MDSCs via caspase-dependent pathways and the Fas/FasL pathway (intrinsic pathway and extrinsic pathways, respectively). The authors also noted that STAT3 inhibition reversed MDSCs suppression on chimeric antigen receptor T (CAR-T) cells in HCC, a new type of targeted therapy for treating tumors [95]. Icaritin was a novel small molecular, which displayed anticancer activities in several cancers [96]. In HCC, Icaritin might exert antitumor roles by modulating the immune microenvironment [97]. It was reported that Icaritin could regulate the dynamics of soluble cytokines and immune checkpoint proteins, mediated by IL-6/JAK/STAT3 pathways in CD8^+^ T cells, neutrophils, macrophages, and MDSCs [98]. Furthermore, Icaritin synergistically enhanced the therapeutic effect of immune checkpoint blockade therapy in HCC mice, showing potential combination values [97]. A single-arm phase I study demonstrated that icaritin had safety profiles and preliminary durable survival benefits in subjects with advanced HCC, associated with immune response and immune biomarkers [99].

### 7.2. Prevention of MDSCs Trafficking

Several attempts have been made to inhibit the migration of MDSCs into tumor sites. Multiple colony-stimulating factor 1 (CSF1/M-CSF) receptor (CSF1R) inhibitors have shown the inhibition of trafficking PMN-MDSCs and M-MDSCs. JNJ-40346527 was a selective CSF1R inhibitor. It could abolish the negative regulation of tumor cell-derived CSF1 on PMN-MDSC recruitment, causing an accumulation of PMN-MDSCs in tumors [100]. The combination of JNJ-40346527 and CXCR2 significantly blocked MDSCs recruitment and reduces tumor growth. Additionally, the addition of anti-PD-1 further improved the accumulation [100].

CCR2 is mainly expressed on specific cell types, such as monocytes, NK, T cells, and MDSCs, particularly for the subset of M-MDSCs [101,102]. CCR2 binds with several ligands (i.e., CCL2, CCL7, CCL8, and CCL12), with the most potent binding preference with CCL2 [101]. The CCL2-CCR2 axis is one of the key players in the trafficking of MDSCs in HCC [22,41]. A study by Wang et al., showed that RS102895, a CCR2 antagonist, could block the chemotactic effect of MDSCs in Hepa1-6-A3B subcutaneous HCC tumors and limit the tumor growth dramatically [41].

The expression levels of CXCR2 and CXCR4 are abundant on MDSCs in both the HCC and non-tumor tissues. Blocking CXCR2 with antagonist SB265610 was observed to suppress MDSCs chemotaxis and facilitate antitumor immunity of CD8^+^ T cells, thus thwarting HCC growth substantially [49]. A phase I/II trial is currently recruiting for CXCR2 antagonist (AZD5069) in patients with metastatic castration resistant prostate cancer (NCT03177187). Another study from Xu et al., proved that MDSCs pretreated with CXCR4 inhibitor AMD3100 dramatically attenuated their ability to migrate to the spleen and liver, indicating that targeting MDSCs recruitment is a novel therapeutic strategy for HCC [42].

### 7.3. Inhibition of Immunosuppression

Considering MDSCs perform remarkable immune suppression functions during carcinogenesis, targeting inhibitory mechanisms might work. Strategies blocking immunosuppression could be carried out mainly through these aspects: alleviating nutrient deprivation, inhibiting ROS production, and helping the functional activation of T cells.

Cyclooxygenase 2 (COX-2) was reported to regulate the expression of prostaglandin E2 (PGE2), which induced the upregulation of ARG1 in MDSCs [103,104]. Thus, the inhibitors of PEG2 and COX2 could impair ARG1 synthesis and the following MDSCs expansion in multiple cancers [105,106,107]. Celecoxib was a selective COX-2 inhibition, reducing COX2 and PEG2 levels in vitro and in vivo in mesothelioma. Dietary celecoxib dramatically decreased the local and systemic expansion of MDSCs in tumor-bearing mice and impeded their suppressive function [105]. It is worth noting that there have been no data yet focusing on the role of COX2 or PEG2 inhibitors on MDSCs in HCC.

In addition to inhibitors targeting COX directly, phosphodiesterase 5 (PDE5) inhibitors could also act as COX analogs. Tadalafil is an FDA-approved PDE5 inhibitor that has been proven to effectively reduce MDSCs, restore T cells in peripheral blood and tumor microenvironments, and has been applied to multiple cancers [108,109]. In HCC, the application of tadalafil enhances the cytokine-induced killer (CIK) cell-based immunotherapy in murine HCC models via suppressing the expansion of MDSCs in an ARG1- and iNOS-dependent manner [110]. Up to now, Tadalafil has been tested in clinical trials of patients with head and neck cancer and multiple myeloma and has shown significant inhibition of MDSCs in patient serum (NCT01697800, NCT01374217).

Some researchers have noticed that histone deacetylase inhibitors (HDACIs) are immune-cytotoxic [111]. Targeting HDACs could regulate the host immune system and enhance cancer immunotherapy [112]. Entinostat is a specific inhibitor of class I HDACs and is approved by the FDA to treat non-small cell lung cancers [112]. Entinostat has been shown to suppressed MDSCs via reducing the activity of Arg-1, iNOS, and COX2 [113]. The combination of entinostat and anti-PD-1 or anti-CTLA-4 has strengthened the inhibitory effects through inhibiting the VEGF, ErbB, and mTOR pathways and the following STAT3 activity in PMN-MDSCs [114]. Entinostat has also been shown to inhibit the migration of MDSCs via interfering with CCR2 [113]. Additionally, entinostat increased cellular sensitivity to TRAIL and triggered tumor-selective death signaling in acute myeloid leukemia [115]. Other HDACIs could also reduce the number of MDSCs through different mechanisms of action and exert antitumor functions, except for vorinostat.

Nitroaspirin has been an immune modulator that could normalize the immune status by correcting the inhibitory enzymatic activities of MDSCs. Orally administered nitroaspirin dramatically reduced the enzymatic activity of ARG1 and iNOS in tumor-bearing mice, which, in turn, inhibited the recruitment and survival of MDSCs and alleviated tumor-specific unresponsiveness [116].

### 7.4. Induction of MDSCs Differentiation and Maturation

Inducing MDSCs maturation into mature myeloid cells is a promising approach. All-trans retinoic acid (ATRA) is a derivative of vitamin A, the deficiency of which caused the expansion of IMCs in mice [117]. ATRA reduced the production of free radicals by stimulating the synthesis of glutathione (a central intracellular antioxidant molecule), thereby promoting the differentiation of IMCs into DC, granulocytes, and monocytes and enhancing antitumor T cells response [118,119]. Considerable evidence has demonstrated that ATRA is an effective compound for HCC therapy and prevention [120]. In addition, ATRA could significantly potentiate the cytotoxic effects of cisplatin. However, it is still unclear whether this enhancement is related to MDSCs [121]. Currently, several clinical trials focus on the combination of ATRA and ICIs, including Atezolizumab, Pembrolizumab, and Ipilimumab (NCT04919369, NCT03200847, and NCT02403778, respectively).

## 8. Perspective and Future

MDSCs are currently considered prominent participants in the development of tumor immune tolerance and valuable prognostic and predictive biomarkers in a variety of tumors. Considering the multiple roles of MDSCs in the tumor microenvironment, targeting them may provide unexpected benefits for HCC patients. In this review, we have summarized some of the mechanisms by which MDSCs regulate the growth and metastasis of HCC through immunosuppression. However, the molecular mechanisms regulating the inhibitory activity of MDSCs in HCC have not been fully elucidated. In addition, the interaction between HCC specific tumor environment and MDSCs needs to be further clarified. Next to immunosuppression, MDSCs are also proved to play many other roles in cancers, such as promoting angiogenesis and forming pre-metastatic niches before tumor metastasis [122]. Whether there are crosstalks between these roles is also an interesting question that is expected to be studied. MDSCs are a highly heterogeneous group, so the impacts and mechanisms of targeting different MDSCs subtypes also remains to be more precisely clarified. With the development of high-throughput technologies in the future, the underlying crosstalk between MDSCs and tumor cells, including genetic, epigenetic, and recognition molecules, will be more fully characterized. As discussed above, monotherapy for MDSCs has shown promising but limited efficacy. The current combination therapy for MDSCs mainly includes immunotherapy, kinase inhibitors, traditional chemotherapeutics, and ATRA. These strategies have improved effectiveness, especially the combination of MDSCs-targeting treatment and immunotherapy. Future research should clarify how much survival benefit the combination therapy can provide and how long its efficacy can last. These problems still need to be confirmed by in-depth preclinical studies and large-scale clinical research in the future.

## Figures and Tables

**Figure 1 cancers-13-05127-f001:**
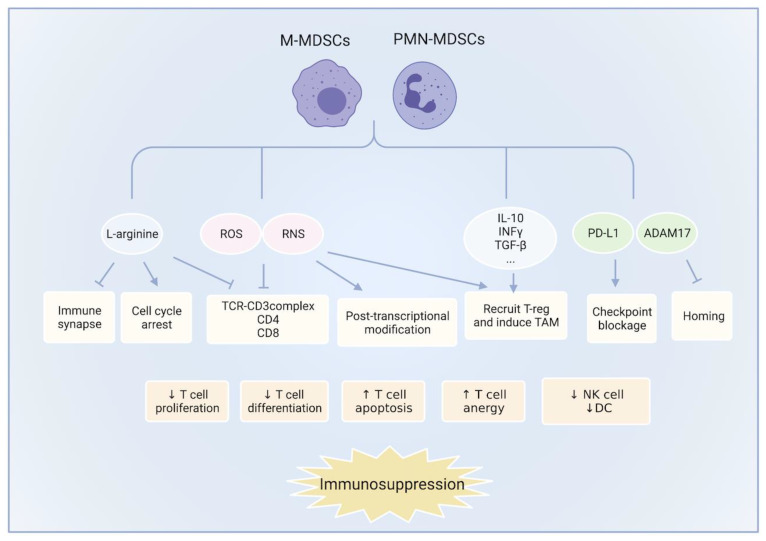
The mechanisms of MDSCs suppress the immune microenvironment: MDSCs execute potent immunosuppressive effects through various mechanisms, including nutrient depletion, oxidative stress, immunosuppressive cell induction, and the expression of PD-L1, AMAD17, and other molecules. These effects inhibit the proliferation and activation of effector cells, such as T cells, NK cells, and DC cells, and promote their apoptosis and anergy, contributing to the formation of an immunosuppressive microenvironment ultimately. The induction of Treg and TAM has also participated in this process. ROS: reactive oxygen; RNS: reactive nitrogen species; Treg: Regulatory T cells; TAM: tumor-associated macrophages; ADAM17: Adisintegrin and metalloproteinase 17; DC: dendritic cell.

**Figure 2 cancers-13-05127-f002:**
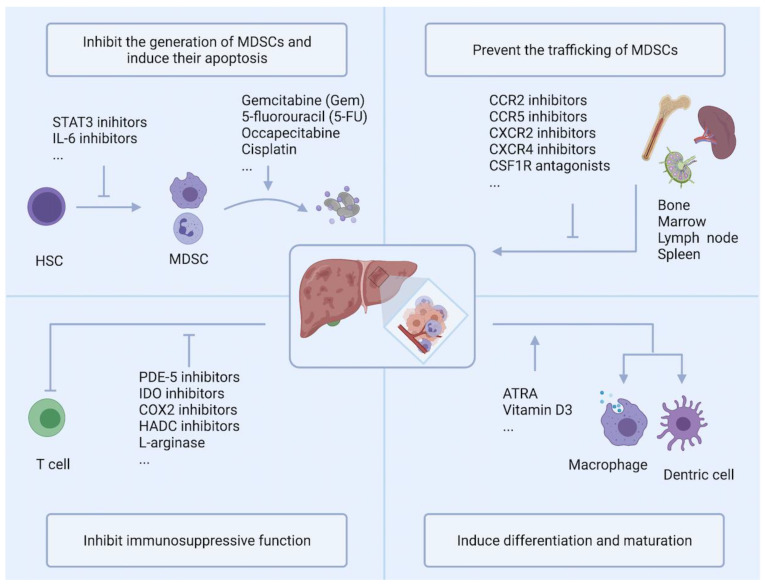
Strategies targeting MDSC can be roughly divided into four categories: Inhibition of MDSCs generation and induction of their apoptosis; Prevention of MDSCs trafficking; Inhibition of immunosuppression; Induction of MDSCs differentiation and maturation. PDE-5: Phosphodiesterase 5; IDO: indoleamine 2,3-dioxygenase; COX2: cyclooxygenase 2; HDAC: histone deacetylase; ATRA: All-trans retinoic acid.

**Table 1 cancers-13-05127-t001:** Signals involved in the activation and recruitment of MDSCs in HCC.

Pathway/Signaling	Phenomena	Intervention strategy	Author/Year	Ref
CCL2	↑ TAMs and MDSCs recruitment into HCC mediated by the interaction between A3B and PRC2.	CCR2 antagonist; Prospect: A3B inhibitor combines with PRC2 complexes inhibitor	Duowei Wang. 2018	[41]
CCRK- IL-6	↑ PMN-MDSC accumulation; ↓ impairing the infiltration of IFNγ^+^ CD8^+^ T cells.	CCRK blockade; IL-6 protein trap; Co-blockade of CCRK and PD-L1.	Jingying Zhou. 2017	[15]
Complement C3	↑ activates MDSCs to produce IL-10 through the P38 MAPK pathway.	Not verified	Ning Wang. 2021	[45]
CXCL1-CXCR2	↑ MDSC accumulation in RIPK3(-) HCC; ↓ CD8^+^ T cells infiltration RIPK3(-) HCC.	CXCR2 antagonist	Yiming Li. 2019	[49]
HIF-1- ENTPD2-Extracellular 5′-AMP	↓ MDSCs differentiate into mature dendritic cells.	ENTPD2 inhibitor	David Kung-Chun Chiu. 2017	[50]
IL-18	↑ MDSCs recruitment in Tlr2^−^/^−^ mice;	TLR2 agonist	Shinan Li. 2015	[51]
IL-1α, IL-1β, and IL-6	The expression of IL-1α, IL-1β, and IL-6 are positively correlated with the proportion of MDSCs and Foxp3+Treg in HCC.	Not verified	Ling Lin. 2021	[48]
IL-6 and GM-CSF	↑ the induction of MDSCs in HCC.	Chemerin (inhibit the expression of IL-6 and GM-CSF)	Y Lin. 2017	[52]
SDF-1α/ CXCR4-IL-6- STAT3	↑ the activation of hepatic stellate cells; ↑ the accumulation and differentiation of MDSCs.	IL-6 antibody; SDF-1α antibody; STAT3 inhibitor; CXCR4 inhibitor;	Y Deng. 2016; Yaping Xu. 2019	[42,43]
COX2 and PEG2	↑ the accumulation of MDSCs in HCC.	COX-2 inhibitor: SC-236	Junru Li. 2018	[46]
IL-21R deficiency	↑ increase MDSC chemotaxis through upregulating chemokines (such as CCL2, CCL7, and CXCL5).	IL-21R knockout	Xinchun Zheng. 2019	[44]
CCL9-CCR1, CCL2-CCR2	↑ the mobilization of PMN-MDSCs in H22 orthotopic hepatoma mice.	Not verified	BaoHua Li. 2020	[22]

A3B: The apolipoprotein B mRNA editing enzyme catalytic polypeptide-like 3B; RIPK3: Receptor-interacting protein kinase 3; HIF-1: Hypoxia-inducible factor-1; Tlr2: Toll-like receptors 2; SDF-1α: Stromal cell-derived factor; COX-2: Cyclooxygenase-2; PGE2: Prostaglandin E2. ↑: increase; ↓: decrease.

**Table 2 cancers-13-05127-t002:** Clinical trials of targeting MDSCs in solid tumors (recruiting, active, or ongoing).

Target	Intervention	Conditions	Phase	Status	Number Enrolled	NCT
Chemotherapy; Anti-PD-1	Gemcitabine + Nivolumab	NSCLC	IV	Recruiting	50	NCT04331626
Chemotherapy; Electrothermal therapy	Gemcitabine; Focused Ultrasound; Gemcitabine + Focused Ultrasound	Breast Cancer; Breast Neoplasms	I	Not yet recruiting	48	NCT04796220
Chemotherapy; Anti-PD-1	Gemcitabine + Nivolumab	NSCLC	IV	Recruiting	50	NCT04331626
Chemotherapy; Anti-VEGF	Capecitabine; Bevacizumab	Glioblastoma	I	Recruiting	12	NCT02669173
Vitamin C; Chemotherapy	Ascorbic Acid + Paclitaxel protein-bound; Cisplatin; Gemcitabine	Pancreatic Cancer; Pancreas Cancer; Pancreatic Adenocarcinoma Resectable; Pancreatic Ductal Adenocarcinoma; Pancreas Metastases	I/II	Active, not recruiting	27	NCT03410030
PDE5 inhibitor	Tadalafil	Astrocytoma	I	Recruiting	16	NCT04757662
TLR9 agonist; Anti-PD-1	CMP-001; Nivolumab	Melanoma; Lymph Node Cancer	II	Active, not recruiting	34	NCT03618641
Anti-PD-1; Diabetes drugs	Pembrolizumb; Metformin	Advanced Melanoma	I	Recruiting	30	NCT03311308
CXCR2 antagonist; androgen receptor antagonist	AZD5069+ Enzalutamide	Metastatic Castration Resistant Prostate Cancer	I/II	Recruiting	86	NCT03177187
CXCR1/2 inhibitor; Anti-PD-1	SX-682; Pembrolizumab	Melanoma Stage III/IV	II	Recruiting	77	NCT03161431
Anti-PD-L1; regional curative treatment	Atezolizumab; stereotactic ablative radiotherapy (SABR)	Metastatic Tumors	II	Recruiting	187	NCT02992912
Vaccinia; Anti-PD-1	Modified Vaccinia Virus Ankara Vaccine Expressing p53; Pembrolizumab	Solid tumors	I	Active, not recruiting	19	NCT02432963
Anti-PD-1 ATRA	Atezolizumab; Tretinoin	Metastatic NSCLC; Recurrent NSCLC	I	Not yet recruiting	18	NCT04919369
Anti-PD-1; ATRA	Pembrolizumab + ATRA	Melanoma Stage III/IV; Advanced Melanoma;	I/II	Active, not recruiting	26	NCT03200847
ATRA; Anti-CTLA4	VESANOID; Ipilimumab	Melanoma	II	Active, not recruiting	10	NCT02403778

Up to now, the combination therapy of MDSCs in liver cancer has been still in the preclinical experimental stage. (https://clinicaltrials.gov. accessed on 9 October 2021).

## Data Availability

The datasets generated and analyzed during the current study are available in the China drug trials repository (https://clinicaltrials.gov accessed on 9 October 2021).

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
