# Peer review of "Therapeutic Values of Myeloid-Derived Suppressor Cells in Hepatocellular Carcinoma: Facts and Hopes"

_cancers, 2021, doi:10.3390/cancers13205127_

Round 1

Reviewer 1 Report

I read with interest the article entitled Therapeutic values of Myeloid-Derived Suppressor Cells in Hepatocellular carcinoma: Facts and Jopes, by wang et al.
The review is exaustive but I have some comments:

Page 3 line 103  specify  where the M-MDSC are in the contexts  of the tumor tissue.

In the  caption of  table n°1 NASH is mentioned  but I have not seen it inside the table.

In consideration  of the numerous abbreviations, I think that is better for the reader, to make a list at the end  of the article.

In paragraph 6.1 specifyng that  the chemotherapy has always been unsuccessful even for  the condition in which  the hepatocellular carcinoma  arises, that is cirrhosis. Chemotherapy often lead to worsening  of liver function.

I have doubt if  table 2 is appropriate for this article (describes other cancers); I would rather  describe  if  the etiology of cirrosis (HBV, HCV  Alcol,  NASH) has an impact of MDSC response.

Author Response

Dear reviewer,

We are delighted to have the opportunity to send you a  point-by-point response and an extensively revised manuscript for our submission. We are resubmitting our manuscript entitled “Therapeutic values of Myeloid-Derived Suppressor Cells in Hepatocellular Carcinoma: Facts and Hopes”(R1). In this revision, we have addressed the questions according to your suggestions. We greatly appreciate the insightful and important comments from you, which have greatly improved the manuscript. We hope that the revised manuscript is now suitable for publication in Cancers.

Yours Sincerely,.

Point-by-point response

Point 1:Page 3 line 103  specify  where the M-MDSC are in the contexts  of the tumor tissue.

Response 1: Thank you for your seggestion. We have rewritten that sentence and added the explaination of  the M-MDSCs distribution.

Point 2/3: In the  caption of  table n°1 NASH is mentioned  but I have not seen it inside the table.In consideration  of the numerous abbreviations, I think that is better for the reader, to make a list at the end of the article.

Response 2/3: We have checked the abbreviation problem in this manuscript and attached an abbreviation list at the end of the review.

Point 4:In paragraph 6.1 specifyng that  the chemotherapy has always been unsuccessful even for  the condition in which  the hepatocellular carcinoma  arises, that is cirrhosis. Chemotherapy often lead to worsening  of liver function.

Response 4: We have added the information about the dam effect of chemotherapy on the liver. This explanation makes the context more convincing.

Point 5:I have doubt if  table 2 is appropriate for this article (describes other cancers); I would rather  describe  if  the etiology of cirrosis (HBV, HCV  Alcol,  NASH) has an impact of MDSC response.

Response 5: Our original table aims to provide the current research status of the combination therapy targeting MDSCs in solid tumors. On the one hand, the application of this type of combination in HCC is limited; on the other hand, we thought that other solid tumor studies might provide a certain reference for the treatment of HCC. Due to time issues, we have not found enough ref to support the influence of the cause of liver cirrhosis on the treatment of MDSCs. If we have the opportunity, we are glad to try again. We are happy to put such interesting content in this review. Thank you for your suggestion!

Reviewer 2 Report

Dear Editor, thank you so much for inviting me to revise this manuscript about hepatocellular carcinoma.

Hepatocellular carcinoma (HCC) remains a frequent malignancy worldwide, representing an important cause of cancer-related death. Modern immunotherapy with immune checkpoint inhibitors (ICIs) has been investigated as front-line treatment in unresectable HCC, as monotherapy or in combination with other anticancer drugs. Several phase III trials exploring the role of ICI-based combinations are ongoing, with the results of these studies still pending and which could further modify the first-line treatment scenario.

Based on these premises, the paper addresses a timely topic.

The manuscript is quite well written and organized.

Tables are comprehensive and clear.

The introduction explains in a clear and coherent manner the background of this study.

We suggest the following modifications:

  • Although the authors correctly included important papers in this setting, we believe a couple of papers should be cited in the introduction (PMID: 33820447; PMID: 28939663), only for a matter of consistency.
  • In addition, the authors should check grammar.
  • Of note, the authors should expand some sections, including a more personal perspective to reflect on. For example, they could answer the following questions – in order to facilitate the understanding of this complex topic to readers: what potential does this topic hold? What are the knowledge gaps and how do researchers tackle them? How do you see this area unfolding in the next 5 years? We think it would be extremely interesting for the readers. 

We believe that major revisions are needed. The main strengths of this paper are that it addresses an interesting and very timely question and provides clear answers, with some limitations. We suggest and the addition of some references for a matter of consistency. 

Author Response

Dear reviewer,

We are delighted to have the opportunity to send you a  point-by-point response and an extensively revised manuscript for our submission. We are resubmitting our manuscript entitled “Therapeutic values of Myeloid-Derived Suppressor Cells in Hepatocellular Carcinoma: Facts and Hopes”(R1). In this revision, we have addressed the questions according to your suggestions. We greatly appreciate the insightful and important comments from you, which have greatly improved the manuscript. We hope that the revised manuscript is now suitable for publication in Cancers.

Yours Sincerely,.

Point-by-point response

Point 1:Although the authors correctly included important papers in this setting, we believe a couple of papers should be cited in the introduction (PMID: 33820447; PMID: 28939663), only for a matter of consistency.

Response 1: Thank you for your suggestion. We have cited these two important papers in the introduction.

Point 2: The authors should check grammar.

Response 2:  Thank you for pointing out our problems. We have checked the grammar of the full text.

Point 3:Of note, the authors should expand some sections, including a more personal perspective to reflect on. For example, they could answer the following questions – in order to facilitate the understanding of this complex topic to readers: what potential does this topic hold? What are the knowledge gaps and how do researchers tackle them? How do you see this area unfolding in the next 5 years? We think it would be extremely interesting for the readers.

 Response 3: Thank you for your valuable suggestions. We have rewritten and expanded "Viewpoint and Future". We think this is a good boost for this review.

Reviewer 3 Report

Dr. Limin Xi et al try to address Myeloid-Derived Suppressor Cells (MDSCs) in HCC. Their attempt is potentially important, however, there are some limitations for publication. 

There is a lack of journal names in the reference section (#2, 25,2, ….).

IFNγ maybe correct in Fig1 and P.7, line24.

They use two different kinds of HSC (hematopoietic stem and hepatic stellate cells). The precise explanations should be described, particulary in P.4, line 183.

May be “Dendric cell” and “HDAC” in Fig2. Please confirm the explanation for the abbreviations.

In P.13, there are two descriptions, “MDSCs can regulate the growth and metastasis of HCC through immunosuppression” and “In addition to immune suppression, it is still unclear whether MDSCs also play other roles in HCC, such as promoting tumor metastasis and causing metabolic disorders”. Please explain them.

Please indicate where readers can check the Table2 in the manuscript.

If MDSCs could work on inflammation in the liver as a tumor-inducing pathophysiology, especially before HCC establishes, its effects on hepatic stellate cells or hepatocytes may be interesting.

Author Response

Dear reviewer,

We are delighted to have the opportunity to send you a  point-by-point response and an extensively revised manuscript for our submission. We are resubmitting our manuscript entitled “Therapeutic values of Myeloid-Derived Suppressor Cells in Hepatocellular Carcinoma: Facts and Hopes”(R1). In this revision, we have addressed the questions according to your suggestions. We greatly appreciate the insightful and important comments from you, which have greatly improved the manuscript. We hope that the revised manuscript is now suitable for publication in Cancers.

Yours Sincerely,.

Point-by-point response

Point 1: There is a lack of journal names in the reference section (#2, 25,2, ….).

Response 1: Thank you for pointing out our incorrect writing. We have corrected the Ref format.

Point 2:  IFNγ maybe correct in Fig1 and P.7, line24

Response 2: Thank you for your comments.We have  modified and unified the “INFγ” in the text, table and in Fig.

Point 3: They use two different kinds of HSC (hematopoietic stem and hepatic stellate cells). The precise explanations should be described, particulary in P.4, line 183. 

Response 3: Thank you for your comments.We have deleted the the abbreviation of hematopoietic stem cell. Furthermore, we listed an abbreviation table at the end of the version.

Point 4: May be “Dendric cell” and “HDAC” in Fig2. Please confirm the explanation for the abbreviations.

Response 4: We have added the abbreviation explansion to both the figure legends and to the article.

Point 5: In P.13, there are two descriptions, “MDSCs can regulate the growth and metastasis of HCC through immunosuppression” and “In addition to immune suppression, it is still unclear whether MDSCs also play other roles in HCC, such as promoting tumor metastasis and causing metabolic disorders”. Please explain them.

Response 5: We have rewritten the “Perspective and future”. To support our description, we quoted a review by Professor  Dmitry I. Gabrilovich (Eur J Immunol. 2010 November; 40(11): 2969–2975. doi:10.1002/eji.201040895).

Point 6:If MDSCs could work on inflammation in the liver as a tumor-inducing pathophysiology, especially before HCC establishes, its effects on hepatic stellate cells or hepatocytes may be interesting.

Response 6: Thank you for your suggestion. This is an interesting point. Due to time issues, We haven't figured out how to write this paragraph yet. If we have the opportunity, we can try to add it to this article.

Round 2

Reviewer 2 Report

The authors modified the paper according to our suggestions. We recommend Acceptance in its current form.

Author Response

.